# Experimental Assessment of Cuff Pressures on the Walls of a Trachea-Like Model Using Force Sensing Resistors: Insights for Patient Management in Intensive Care Unit Settings

**DOI:** 10.3390/s22020697

**Published:** 2022-01-17

**Authors:** Antonino Crivello, Mario Milazzo, Davide La Rosa, Giacomo Fiacchini, Serena Danti, Fabio Guarracino, Stefano Berrettini, Luca Bruschini

**Affiliations:** 1ISTI-CNR, Institute of Information Science and Technologies, 56124 Pisa, Italy; antonino.crivello@isti.cnr.it (A.C.); davide.larosa@isti.cnr.it (D.L.R.); 2The BioRobotics Institute, Scuola Superiore Sant’Anna, 56025 Pontedera, Italy; serena.danti@unipi.it; 3Otolaryngology, Audiology and Phoniatric Operative Unit, Department of Surgical, Medical and Molecular Pathology and Critical Care Medicine, University of Pisa, 56123 Pisa, Italy; giacomo.fiacchini@unipi.it (G.F.); stefano.berrettini@unipi.it (S.B.); luca.bruschini@unipi.it (L.B.); 4Department of Civil and Industrial Engineering, University of Pisa, 56122 Pisa, Italy; 5Cardiothoracic and Vascular Anaesthesia and Intensive Care, Department of Anaesthesia and Critical Care Medicine, Azienda Ospedaliero Universitaria Pisana, 56123 Pisa, Italy; f.guarracino@ao-pisa.toscana.it

**Keywords:** COVID-19, intubation, tracheoesophageal fistula, tracheal lesions, acute respiratory distress syndrome, modeling, intensive care unit

## Abstract

The COVID-19 outbreak has increased the incidence of tracheal lesions in patients who underwent invasive mechanical ventilation. We measured the pressure exerted by the cuff on the walls of a test bench mimicking the laryngotracheal tract. The test bench was designed to acquire the pressure exerted by endotracheal tube cuffs inflated inside an artificial model of a human trachea. The experimental protocol consisted of measuring pressure values before and after applying a maneuver on two types of endotracheal tubes placed in two mock-ups resembling two different sized tracheal tracts. Increasing pressure values were used to inflate the cuff and the pressures were recorded in two different body positions. The recorded pressure increased proportionally to the input pressure. Moreover, the pressure values measured when using the non-armored (NA) tube were usually higher than those recorded when using the armored (A) tube. A periodic check of the cuff pressure upon changing the body position and/or when performing maneuvers on the tube appears to be necessary to prevent a pressure increase on the tracheal wall. In addition, in our model, the cuff of the A tube gave a more stable output pressure on the tracheal wall than that of the NA tube.

## 1. Introduction

The coronavirus disease 2019 (COVID-19) outbreak has raised many critical issues in the management of patients affected by acute respiratory distress syndrome (ARDS) in an intensive care unit (ICU) setting [1,2]. Among others, the high incidence of full-thickness tracheal lesions (FTTLs) and tracheoesophageal fistulas (TEFs), and their potential life-threatening complications, such as pneumomediastinum, pneumothorax, and subcutaneous emphysema, have been reported in patients who underwent invasive mechanical ventilation (MV) [3,4]. This procedure consists of ventilating the respiratory apparatus via an endotracheal polymeric tube with an inflatable cuff that seals the tracheal duct. Depending on the targeted application, the endotracheal tube may have an embedded reinforced metal coil to stiffen the structure, making it less likely to be obstructed [5]. However, independent of the tube type, a cuff pressure ranging between 20 and 30 cmH_2_O is always recommended to avoid damage or trauma to the host tissue [6,7].

Many etiopathogenetic hypotheses have been proposed to explain the unprecedented increase in complications observed in ARDS patients treated with MV [3,8]; however, to date, a clear explanation has not been found.

One proposed mechanism relies on performing the invasive MV in the prone (P) position with the patient’s head laterally rotated. Specifically, by moving the patient from the supine (S) to the P position, the orotracheal tube cuff is supposed to increase its pressure on the tracheal wall, causing tissue lesions. However, there are no studies demonstrating how these lesions are formed at a laryngotracheal level in this body position.

We designed an experimental study to measure the pressure exerted by the cuff on the wall of a test bench mimicking the laryngotracheal tract, placed in different orientations (S and P), with different loading configurations (torsion and bending of the orotracheal tube). The experimental protocol considered both types of endotracheal tubes, namely armored (A) vs. non-armored (NA), under a progressive increase in the internal cuff pressure.

Understanding the causes of FTTL and TEF formation in patients affected by ARDS would greatly improve patient management in the ICU, which can result in faster and less complicated recoveries than those currently experienced.

## 2. Materials and Methods

The test bench was designed to acquire the pressure exerted by endotracheal tube cuffs inflated inside a cylindrical pipe, simulating an artificial model of a human trachea during the intubation maneuvers.

The endotracheal tubes used in this study were armored (A) (Unomedical UM61214075, 7.5 mm, made by ConvaTec, Deeside, UK) and non-armored (NA) (ETT-P22-75, 7.5 mm, made by Medis Medical, Tianjin, China). The distinctive feature of the A tube is the metal wire coil embedded in its wall, which keeps the lumen of the tube open when it is bent. Moreover, the A tube is more flexible than the NA tube, so it is less prone to kink and/or to being occluded when bent. Because the NA tube is pre-formed and more rigid than the A tube, it does not require the use of a stylet for a successful intubation. In contrast, the A tube, being flexible, always requires the use of a stylet. The artificial laryngotracheal tract used in this study was built using a corrugated plastic tube to replicate the characteristics of the larynx and trachea. To monitor the endotracheal tube cuff pressure exerted on the tracheal segment, we deployed four force sensing resistors (FSRs) along the tracheal tract, positioned on the four sides of the internal tube (i.e., 0°, 90°, 180°, and 270°, clockwise). Figure 1 shows a sketch of the prototype from the front and from a longitudinal cross-section perspective. To assure ease of repeatability and high-precision measurements, the tracheal segment of the prototype where the FSRs were located was made of a transparent material (i.e., plexiglass).

The sensors used to measure and record the pressure generated by the inflated cuffs are called FSRs. They are made of a conductive polymer that changes its electrical resistance proportionate to the force applied to its surface. These sensors have been widely applied to acquire different human body functional parameters, such as foot pressure [9], respiration rate [10], finger forces [11], muscle activity [12], and body movements during sleep [13]. The product used in this study was an Interlink FSR^®^ Model 400, which is characterized by its thinness (i.e., 0.3 mm) and circular active area of ~5 mm diameter. These features allow the sensors to be positioned and fastened appropriately onto the inner surface of the tube without creating additional thickness or deformation that may interfere with the endotracheal tube maneuvers, thus influencing the measurements. Figure 2 shows the main dimensions of an FSR and its resistance vs. force characteristics.

The FSRs were connected to a Raspberry Pi single-board computer through multiple microchip MCP3424 analog-to-digital converter (ADC) modules, which translate the raw input voltage from the sensors to a digital format data readable by a software application. The MCP3424 is an 18 bit, 4-channel delta-sigma ADC with differential inputs, self-calibration of internal offset, and gain at each conversion. It also provides an internal programmable gain amplifier, an internal voltage reference (i.e., 2.048 V ± 0.05%, 15 ppm/°C), and a programmable data rate of up to 240 samples per second. The ADC modules communicate with the Raspberry Pi board through the I2C bus in high-speed mode (i.e., 3.4 MHz).

A dedicated application run on the Raspberry Pi oversaw sensor data collection through a sampling frequency of 3 Hz and subsequently stored it in a local file before transmitting it on a remote computer for visualization. The acquired sensor voltage data were analyzed with MATLAB (v2019b), converted into pressure values (in kilopascals), and plotted in real time to provide immediate feedback during the maneuvers.

The FSRs, although versatile, compact, and cheap, are subject to physical effects due to their construction, which necessitates careful calibration before usage to reduce sensing errors. Specifically, it is recommended that the calibration phase of each sensor be conducted in an environment that is close to the final application [15]. Remarkably, FSRs show a power law behavior within the force range from 0 to 20 N. However, the 0−4 N range shows mostly a non-linear relation, thus highlighting the need for accurate calibration of the FSRs in low-pressure range setups [16]. The FSRs were experimentally characterized to obtain the actual response curve, considering the effect of the ADC conditioning circuit. Briefly, 8 sensors were tested by applying a sequence of calibration weights (i.e., 10, 20, 50, 100, and 200 g). After applying each weight, the sensor output was left to stabilize before noting the reading. For each weight, 10 measures were taken to average the outcome. From the experimental readings, we designed the fitting curve using a cubic polynomial regression (Figure 3). Lastly, we computed the inverse function that was used to estimate the force applied on the sensor surface.

### 2.1. Measurement Protocol

Pressure values were measured before and after applying a maneuver on two types of endotracheal tubes (NA/A) placed inside two mock-ups resembling the pharyngeal, laryngeal, and tracheal tracts. The mock-ups had two different diameters (i.e., 20 and 25 mm). Increasing pressure values were used to inflate the cuff (30/40/50 cmH_2_O). Even though the recommendations suggest limiting cuff pressure to below 30 cmH_2_O, we decided to slightly increase this value in order to consider the potential for operator error. For each configuration of the tube-test bench, we inserted the tube with the curvature shown in Figure 1 to replicate the S position of the patient. Then, we verified that the cuff reached the correct position, namely with the middle section of the cuff in contact with the four sensors. We inflated the cuff with the targeted pressure, let the sensor signals stabilize, and then noted the pressure values obtained from each sensor. Before applying a different maneuver, we deflated the cuff, recovered the original geometrical configuration, and reinflated the cuff at the target pressure. Thereafter, we applied two different maneuvers (i.e., torsion (T) and bending (B)) with amplitudes equal to 90°, along two different directions (i.e., clockwise and counterclockwise). To perform the experiments with the test bench replicating the P position, the same set of experiments were performed with the test bench rotated by 180° along the median axis of the mock-up.

### 2.2. Statistics

We clustered pressure values before and after maneuvers based on the following group types: input pressures (30/40/50 cmH_2_O), body position (S/P), maneuver (T/B), and type of endotracheal tube (NA/A). Data from the four sensors were averaged at each measurement. Data processing was carried out using jamovi software (V1.6.16.0).

The mean and standard deviation (SD) for the pressure values before the maneuvers were evaluated. We grouped data from both the torsion and bending maneuvers, giving a comprehensive picture of the initial configuration, before the maneuvers, across input pressures. Later, we performed independent Student’s *t*-test analyses to evaluate the statistical differences, if any, between the groups (NA/A and S/P) at fixed input pressures. The *p*-value threshold was set at α equal to 0.05.

The differences between the pressure values after and before applying a maneuver were also analyzed. In this case, an analysis separating the outcomes from each type of maneuver was used, gathering the values from the two opposite directions used for each maneuver (T/B). Thus, we performed independent *t*-test analyses to evaluate the statistical differences, if any, between the groups (NA/A and S/P) at fixed input pressures, for each type of maneuver (i.e., T/B). The *p*-value threshold was fixed at α equal to 0.05. In all tests, the null hypothesis concerned the lack of statistical difference between mean values.

## 3. Results

The dataset was tested to check the normality, which was later confirmed using the Levene’s Test. Statistical analyses concerning the pressure values at the initial configuration are reported in Figure 4. Two different representations of the dataset with boxplots are presented, separately comparing the outcomes grouped by the body position (S vs. P) and type of tube (NA vs. A) against the diameter of the trachea (20 mm vs. 25 mm) and input pressure (30–50 cmH_2_O). Figure 4A–D shows the proportional increase in the measured pressure against the input pressure using the S vs. P classification. Moreover, significant statistical differences between mean values, independent of input pressures and tracheal diameter, were observed. We noticed that the pressure values measured when using the NA tube were usually higher than those measured with the A tube. This effect is particularly evident in Figure 4B, which shows the results for the prone position with a 20 mm trachea, for which the pressure amplitude reached ~6 kPa. In the 25 mm trachea (Figure 4C,D,G,H), the mock-up was not able to detect any relevant pressure (i.e., contact) in 30–40 cmH_2_O. Measurable outcomes were obtained only at 50 cmH_2_O, when the cuff was actually compressed. These results are similar to those of the 20 mm tracheal mock-up, although with amplitudes of ~2 kPa for the NA tube and almost negligible for the A tube.

Figure 4E–H shows the dataset using a grouping based on the body position (S vs. P). In this case, we had a different scenario for the 20 mm trachea: when increasing the input pressures, the mean values tended to decrease in their significant statistical difference. Focusing on Figure 4E (NA tube—20 mm trachea), at 30 cmH_2_O, we estimated a *p*-value below 0.001 but, already at 40 cmH_2_O, the mean values were statistically similar. Concerning the A tube, this effect occurred only at 50 cmH_2_O.

By changing the organization of the dataset, we used as a dependent variable the difference in the pressure values before and after performing a maneuver. We compared the effects of two different maneuvers (i.e., T and B) on the tracheal pressures. Figure 5A–D shows the results obtained using a grouping based on the type of tube (NA vs. A), while Figure 5E–H shows the body position (S vs. P).

The comparison between NA and A tubes displays significant differences between the mean values independent of the type of maneuver and input pressure, with amplitudes higher for the NA tube than those associated with the A tube. In contrast, an increasing trend of the values with the input pressure was not detected. As also shown in Figure 4, no relevant pressures were observed using a 25 mm trachea with input pressures below 50 cmH_2_O.

A similar scenario was observed by applying the classification based on the body position (Figure 5E–H), even though the statistical differences between mean values were not confirmed in all cases. Specifically, this occurred in relation to the mean values associated with the bending maneuvers at 40 cmH_2_O for the 20 mm trachea with the NA tube, torsion maneuvers at 50 cmH_2_O for the 20 mm trachea with the A tube, and bending maneuvers at 50 cmH_2_O for the 25 mm trachea with the NA tube.

## 4. Discussion

We investigated the effects of pressure exerted by endotracheal tubes in a mock-up resembling the laryngotracheal tract. Our study aimed to unveil the mechanisms that induce trauma on the tracheal segment of patients suffering from ARDS who were treated with MV. We used two endotracheal tube types (i.e., NA/A) and observed the results by reference to body configuration (i.e., S vs. P) and pressure variations in the cuff. International guidelines recommend keeping the endotracheal tube cuff pressure between 20 and 30 cmH_2_O [6,7] to avoid serious complications such as aspiration pneumonia, tracheal ischemia, FTTLs, and TEFs [17,18]. Furthermore, we used values up to 50 cmH_2_O to investigate the effect of possible human error. Usually, this pressure is periodically monitored via devices connected to the cuff pilot balloon, which has a mechanical valve that prevents any oxygen leakage [19]. However, the cause–effect relationship of the mechanical pressure on the tracheal tract is still unclear, in particular during the maneuvers exerted on the endotracheal tube or on the patient’s body. This issue has become remarkably relevant during the COVID-19 pandemic, as a large number of patients have been treated with prolonged MV in different body configurations. Marti et al. published an in vitro study on the deflation of cuffs placed in a test bench over time [20]; however, to the best of our knowledge, our investigation is the first assessment of the mechanical pressure exerted by a cuff in a simulated environment. The underlying hypothesis of our study is the variation in the pressure exerted by the cuff on the tracheal wall depending on the type of tube (NA/A), the maneuvers performed on the tube (T/B), and on the patient’s body position (i.e., S/P), against the same initial insufflation pressure.

The first result concerned the pressure measurements obtained after inserting the tube in the model and before applying the maneuvers. Pressure values measured when using the NA tube were higher than those obtained using the A tube, especially in the P position (Figure 5A–D). This is probably due to the superior stiffness of the NA tube, which exhibited a higher preformed curvature than the A tube. These differences may be compensated in vivo by the viscoelastic behavior of the NA tube polymer at the body temperature, which was not included in our experimental set-up. Therefore, we can consider the measured pressures as instantaneous values that may occur just after the positioning of the tube in the laryngotracheal tube. It is reasonable to hypothesize that the A tube, being more flexible than the NA tube, would require fewer corrections of the cuff. Instead, the NA tube might require careful placement and tuning, especially in the initial period after intubation and before the patient’s body temperature induces a shape variation. Another interesting difference between the NA and A tubes was the shape of the cuffs once inflated. The NA cuff took the form of an ellipsoid whereas the A cuff took the form of a regular cylinder. The contact of the surfaces with different curvatures may be another factor affecting the exerted pressures and their distribution on the tracheal wall. This deserves a dedicated investigation.

Another statistically significant result concerned the difference identified in the 20 mm mock-up with the NA tube and the 30 cmH_2_O insufflation pressure from the S to P position, which was not observable for higher insufflation pressures and for the A tube (Figure 4E–H). This is probably attributable to the stiffness of the NA tube and to the maneuvers on the external connector of the endotracheal tube during the S–P maneuvers. As mentioned above, this was not evident for the A tube because of its flexibility.

Using the 20 mm mock-up, we also observed significant differences in output pressure when using either the NA or A tube, independent of the type of maneuver (torsion vs. bending) and input pressure, with amplitudes associated with the NA tube being higher than those of the A tube (Figure 4A–D). In our opinion, this is also due to the different flexibility of the endotracheal tubes. Therefore, it is reasonable to recommend that careful tuning and monitoring of cuff pressure should be performed after each maneuver on the endotracheal tube, especially if the NA tube is employed. Body position appeared not to consistently influence the pressure on the tracheal wall when applying torsion and bending maneuvers on the endotracheal tube, regardless of the type of tube and the insufflation pressure used (Figure 4E–H).

In contrast, the results from the 25 mm mock-up were less informative. This is due to the exact nominal dimensions of the inflated cuff and the trachea-like structure producing reduced contact. As a result, contact occurred only at the highest insufflation pressures. This simple deduction opens up an interesting discussion as to the applicability of the most common endotracheal tubes that have a cuff expandable up to 25 mm. The tracheal segment has a highly variable diameter of 10−27 mm [21]. Therefore, the general employment of a 25 mm cuff tube with the recommended input pressure may be either dangerous or inefficient. As a consequence, a preliminary evaluation of the diameter of the tracheal segment (e.g., using parameters such as the weight/height ratio of the patient) should be conducted to fine-tune the input pressure or, if available, to inform the use of an endotracheal tube with a larger cuff diameter. However, it is important to stress that, at this point in time, a scaling factor for input pressures against the diameter of the tracheal segment is not available.

This study has three main limitations. The first is intrinsic to all studies performed on experimental models, as they cannot perfectly replicate the in vivo conditions. In particular, the use of a thermostatic chamber could have obviated the permanent stiffness of the NA endotracheal tube and, with its use, we could have verified the change in cuff pressure exerted on the tracheal wall over time. However, viscous phenomena in polymeric materials are not immediate and we were interested in assessing the pressure values immediately upon insertion in order to evaluate the mechanical effect on the trachea-like wall. A future study will include a thermo-controlled room in which a mechanical assessment will be performed, in order to highlight the contribution of body temperature. The second aspect is the lack of a real tracheal epithelium and endotracheal secretions enabling the sliding of the cuff on the tracheal walls during torsion and bending maneuvers, which could provide insight into the tribological phenomena concurring or preventing damage mechanisms over time. While this limitation cannot be overcome with our current mock-ups, a dedicated study involving ex vivo tissues may help to assess these phenomena. Finally, an improvement in the evaluation of the cuff pressure, over time, is needed to investigate the relaxation phenomena and the effects of maneuvers over time. This would mimic the condition of the current approach for treating COVID-19 patients but, also, in this case, an ex vivo model would better reproduce the tribological mechanisms of the involved tissues.

In addition to the abovementioned limitations, our study provides some relevant take-away messages on the application of endotracheal tubes for MV. In particular, the first important conclusion concerns the need for periodic checks of the cuff pressure upon changing the body position and/or performing maneuvers on the tube. The latter occurs regularly in daily practice even simply by rotating or hyperextending the patient’s head. Moreover, the cuff of A tubes appeared to give more stable output pressures on tracheal walls than those of NA tubes. Therefore, instead of the common practice of ICU personnel to use NA tubes, the use of A tubes should be considered. In the specific case of our tertiary referral hospital, patients coming from the operating room intubated with an A tube are promptly reintubated with an NA tube. In the current literature, scientific articles justifying this clinical practice are missing. It seems that the practice has originated from practical experience gained in the field.

## 5. Conclusions

The outbreak of the COVID-19 pandemic has brought new attention to the well-known practice of ventilating patients affected by respiratory tract pathologies. To the best of our knowledge, our study delivers, for the first time, an investigation on the loads exerted by the contact of endotracheal tube cuffs on the laryngotracheal tract. We used a mock-up of the anatomic system with two different transversal sizes and two different designs of endotracheal tubes (i.e., armored vs. non-armored) to assess the effect of each device when using specific maneuvers and loads, characteristic of common practices. Despite the intrinsic limitations of the model, we unveiled a number of interesting findings. The most important outcome for clinicians concerns the superior wall pressures induced by NA tube cuffs due to their specific design. This effect is more significant for patients placed in a prone position despite the inlet pressure being kept at 30 cmH_2_O. Interestingly, A tubes induced more stable wall pressures than those produced by NA tubes. Another interesting point to note relates to the transversal dimension of the laryngotracheal tract. Current procedures and tube designs do not consider anatomical differences among patients, notwithstanding that wall pressures may vary significantly depending on the actual dimensions of the cross-sectional diameter.

In conclusion, although we concede that current clinical practices have not resulted in frequent complications, based on recent scientific evidence, our findings support a reconsideration of the current approach to tracheal intubation aimed at MV in ICU patients. Specifically, A tubes should be preferred to NA tubes, and face-down pillows with a central hole to pass NA tubes should be used when pronation is required.

In view of this, since a full understanding of the damage mechanisms is still missing, we think that future studies in this field should investigate in detail such aspects of MV practices, through both ex vivo and in vivo approaches, to improve patient care.

## Figures and Tables

**Figure 1 sensors-22-00697-f001:**
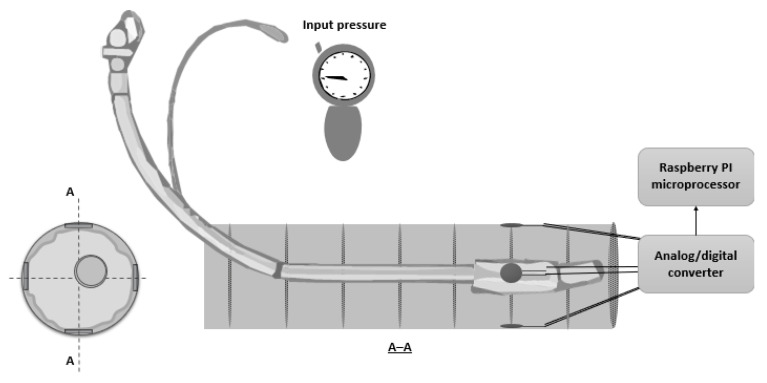
Front view and longitudinal cross-section of the prototype.

**Figure 2 sensors-22-00697-f002:**
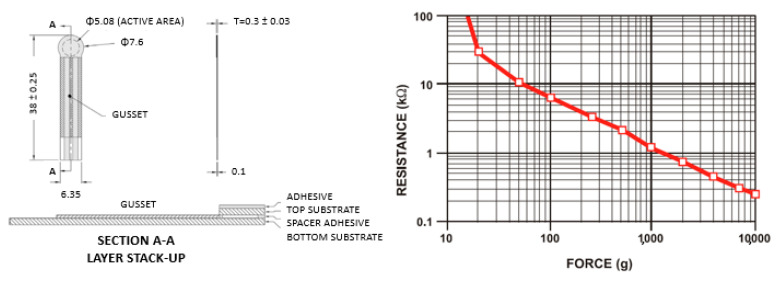
Interlink FSR^®^ Model 400 mechanical data (**left**) and typical force vs. resistance response (**right**) from the datasheet [14].

**Figure 3 sensors-22-00697-f003:**
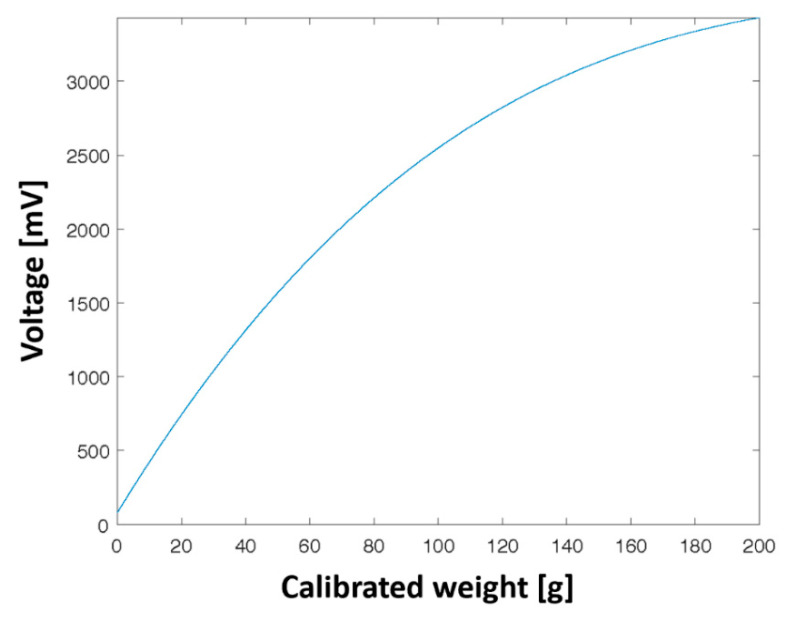
FSRs experimental characterization curve expressed as applied calibrated weight vs. ADC input voltage.

**Figure 4 sensors-22-00697-f004:**
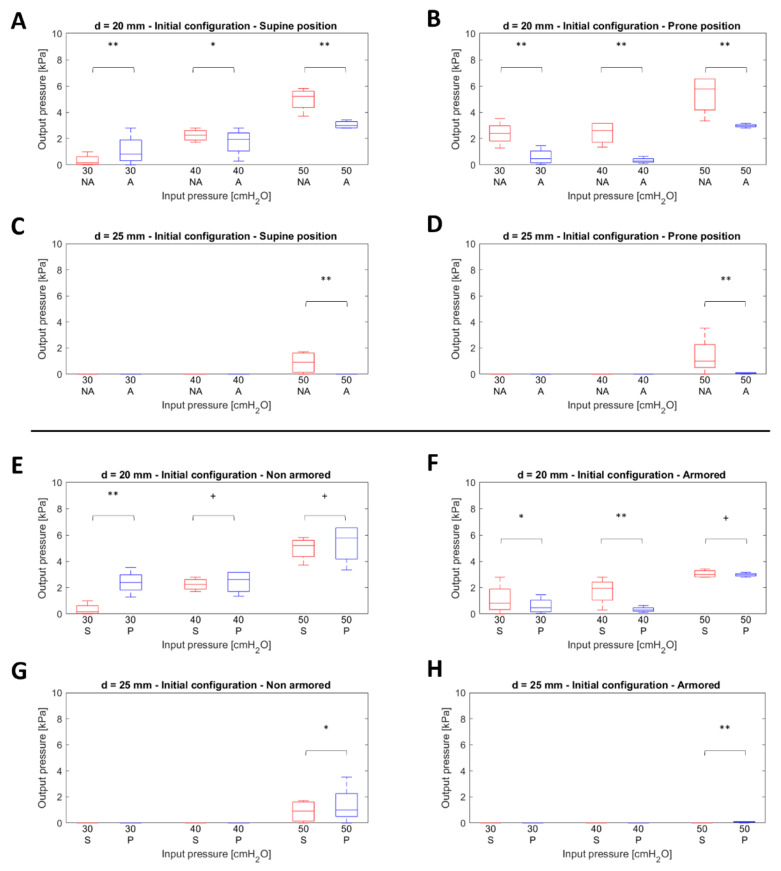
Statistical analysis to determine the differences in pressure, if any, before performing a maneuver. (**A**–**D**) Box plots showing the comparison between non-armored (NA) and armored (A) tubes across diameters of the trachea and position (S—supine, P—prone). (**E**–**H**) Box plots showing the comparison between body position (S/P) against diameters and type of endotracheal tube (NA/A). Legend: ** *p* < 0.001, * 0.002 < *p* ≤ 0.05, and + *p* > 0.05.

**Figure 5 sensors-22-00697-f005:**
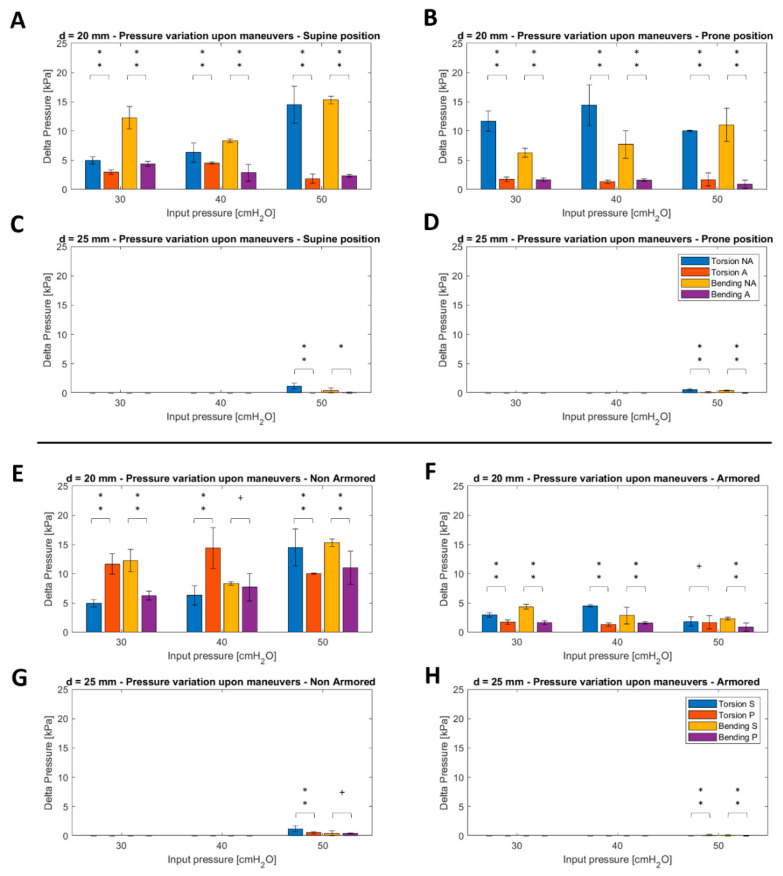
Statistical analysis to determine the differences in pressure variations, if any, after performing a maneuver. (**A**–**D**) Histograms showing a comparison between non-armored (NA) and armored (A) tubes across diameters of the trachea and position (S—supine, P—prone). (**E**–**H**) Histograms showing the comparison between body position (S/P) across diameters and type of endotracheal tube (NA/A). Legend: ** *p* < 0.001, * 0.002 < *p* ≤ 0.05, and + *p* > 0.05.

## Data Availability

Data are available upon request to the corresponding author.

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
