# Peer review of "Experimental Assessment of Cuff Pressures on the Walls of a Trachea-Like Model Using Force Sensing Resistors: Insights for Patient Management in Intensive Care Unit Settings"

_sensors, 2022, doi:10.3390/s22020697_

Round 1

Reviewer 1 Report

Thank you for the opportunity to review this interesting manuscript. In this experimental study, authors measured pressures on the walls of trachea-like model in different scenarios with the use of two different tubes. They found non-armored tubes caused higher pressure values than the armored tubes. I think the manuscript is informative and well written. I have only minor comments, suggestions:

  1. I think the most important information (for a clinician) is that even if the cuff pressure is kept at 30 cm H2O the wall pressure increases in proned patients with NA tubes. Could you please highlight/stress out this information more in the manuscript?
  2. In the conclusion, you recommend periodic checks of the cuff pressure upon changing body position. I’m sure this is important to prevent high cuff pressures. But, in fact, based on your data, this is not going to change the increased tracheal wall pressure (even if the cuff pressure is low).
  3. In the conclusion, “the common practice could be reconsidered towards the use of NA tubes” Why? You have shown A tubes are better, right?
  4. So what should a clinician do? Should armed tubes be used in those patients who are to be proned? Should face down pillows be used in those patients with NA tubes, to prevent head rotation?

Reviewer 2 Report

Overall, this is a well-done and original research effort that touches on a contemporary issue of importance in medical care related to long-term intubation of patients, and the compromise to care by endotracheal cuff lesions. With minor modifications this would be a good addition to the readership.

The paper presents a methodical and comprehensive approach to a bench test to explore the issues of pressure generated on the trachea as a result of intubation. The paper is informative and complete, but it is recommended the authors kindly consider the following comments and observations:

  • The significance of armored versus unarmored (standard) endotracheal tubes assumes the reader is aware of the significant differences that accompany intubation. The armored will have a curve and have a preset curvature, and the unarmored are much easier to bent, but more difficult to insert and thus may need a stylet for successful intubation. Some mention of the implications will hep the reader more fully appreciate the work.
  • Similar to the point above, the tests on bending and torsion and the link to physiology or operational significance could be expanded. To mention, for instance, what conditions torsion of the ETT occurs would help the reader pace the work in context.
  • What percentage of intubated patients are prone versus supine? The paper discusses this scenario, and it would help to better understand the significance and compromise in care. The transition is presumably when the ETT undergoes deformation/change/rotation.
  • Figure 2 is very informative, but the left-hand side of the figure is very small and very difficult to read, so it adds very little value. Please expand or provide a clearer drawing.
  • The paper uses corollaries to pressure in the form of resistance, forces, and pressure (itself), but the use of units is not consistent or translated for the reader. For instance, Figure 2 uses force and resistance, but the narrative speaks in terms of voltage.  Indeed, the scale for Force is in grams, but the narrative mentions Newtons, and it is not clear if the force measurements are for Figures 2 or 3, or what curve-fit has been performed – adding the equation would be helpful to the reading to link the discussion. What is the significant difference between Figures 2 and 3 – the reader must dig into the narrative and infer in order to extract the key message.
  • Results are discussed in kPa, but is not clear if Figure 3 is the source of the calibration.
  • The resistance of the cuff is significance and it is implied that the difference between input pressure and output pressure is a result of the materials and construction of the ETT.
  • It might be helpful to comment on the curvature of the armored cuff and the way in which results are impacted by the residual resistance to change it might offer.
  • The discussion is highly descriptive with a LOT of very good information, but it would help the reader to summarize the significance of the discussion or outline the course of the narrative – as is, the reader gets slightly lost with the amount of information.
